# Assessment of the quality attributes and oxidative stability of fish balls with the addition of fig powder during frozen storage

Md. Sakib Hasan[1], Md. Nowshad Mahmud Choyon[1], Md. Numan Islam[2], Md. Golam Rabby[1], Ananya Raiyan[1], Md. Mohaiminul Islam[1], Nishat Tabassum Prokrite[1], Md. Abid Hassan[1], Nishat Chadni Liza[1], Suvasish Das Shuvo[1], Rashida Parvin[1]*, Md. Ashrafuzzaman Zahid🅖[1]*

1 Department of Nutrition and Food Technology, Jashore University of Science and Technology, Jashore, Bangladesh, 2 Department of Food Science and Technology, University of Nebraska Lincoln, Lincoln, Nebraska, United States of America

* ashraf@just.edu.bd (MAZ); rashida.nft@just.edu.bd (RP)

## Abstract

This study evaluated the oxidative stability and physicochemical attributes of fish balls made from *Pangasius pangasius* (Pangus) by the addition of natural antioxidant *Ficus carica* (Fig fruit) and other synthetic antioxidants, Butylated hydroxytoluene (BHT) and Ascorbic Acid (AA), and the fish balls were stored under freezing conditions for two months. Four different types of fish balls were made with or without antioxidants: 1) T0 (without antioxidants); 2) T1 (0.02% BHT); 3) T2 (0.05% AA); 4) T3 (1% fig powder). Incorporating antioxidants significantly retained ($p < 0.05$) lower TBARS, pH, and yellowness while holding higher DPPH and redness values throughout storage. The addition of fig powder (FP) significantly ($p < 0.05$) reduced oxidation, free radical scavenging activity, and pH values. The fig's antioxidant potential was non-significantly ($p > 0.05$) different from BHT and AA. Moreover, the fig retained a significantly higher moisture content ($p < 0.05$), essential for maintaining fish ball quality. Furthermore, the cooking loss increased non-significantly ($p > 0.05$) among treated samples due to lower moisture and fat loss. Sensory assessment regarding color values of all treated and non-treated samples, FP showed a significantly ($p < 0.05$) higher redness value and a significantly ($p < 0.05$) lower yellowness when compared to control samples throughout the storage period. Therefore, adding FP as a natural antioxidant could reduce oxidation and maintain the others physicochemical characteristics of fish balls under freezing conditions.

## 1. Introduction

Fish balls are one of the popular ready-to-eat fish-based products. It is generally prepared by deboning fish and heating it to remove fishy odour [1]. Fish balls and other

**Data availability statement:** All relevant data are within the manuscript and its Supporting information files.

**Funding:** This research received a grant from the University Grant Commission of Bangladesh.

**Competing interests:** The authors have declared that no competing interests exist.

surimi-based products are consumed as snacks due to their distinct flavour and taste globally [2]. China's surimi-based product business produced 1.334 million tons in 2021, with Fujian Province accounting for almost one-third of that total. The apparent consumption was 1.2219 million tons, and the market size reached CNY 18.732 billion, a 12% increase over 2020 [3]. Fish-based products, especially fish balls, have gained more attention as essential commodities for human consumption globally due to their unique appearance, nutritional value, and low cholesterol, essential vitamins, minerals, and polyunsaturated fatty acids [2,4–6]. Underutilized freshwater and marine fish species have traditionally been used to make fish balls [7]. Pangas (*Pangasianodon Hypophthalmus*), a farmed growing freshwater fish source, gained popularity worldwide [8]. This fish gained popularity due to its rich sources of nutrition, low price, digestibility, and white meat, which might be the probable reasons for its utilization in fish ball preparation [8,9]. Even though fish balls and other fish products are a rich source of nutrition, especially protein, water, and unsaturated fatty acids, they are vulnerable to oxidation and enzymatic and non-enzymatic rancidity during the storage period [10,11]. Therefore, adding preservatives is essential to enhance shelf life, reduce oxidation of protein and lipids, and maintain sensory attributes and nutritional value [12].

Common synthetic antioxidants, including BHT, BHA, ascorbic acid (AA), and propyl gallate, are used in the meat and fish industries to control and purify free radical production and peroxyl radicals. Applying these synthetic antioxidants preserves the quality of meat and fish products by regulating lipid and protein oxidation [13,14]. However, synthetic antioxidants can potentially increase shelf life; they show their adverse effects by indicating their harmfulness, including toxicity and carcinogenicity [15,16]. Several diseases like allergies, diabetes, and increased cancer risk are associated with the use of synthetic preservatives in foods [16]. Current safety concerns regarding the use of natural preservatives instead of synthetic ones, studies have demonstrated the benefits of using natural antioxidants from various fruits and vegetables or their by-products to retain food quality and minimize harmful effects [17–20].

*Ficus carica* (Figs) is one of the most populous species belonging to the *Moraceae* family [21,22]. It seeks public attention as a fruit in both dry and fresh forms, especially in the Mediterranean diet [23,24]. It contains abundant fibre, vitamins, minerals, amino acids, organic acids, and sugars [24–26]. Furthermore, figs are potential sources of bioactive compounds, including cyanidin, rutin, chlorogenic acid, Catechin, and luteolin [24,27]. These bioactive compounds play a significant role due to their antioxidant potential, especially scavenging free radicals and quenching singlet oxygen, antimicrobial, and anti-inflammatory properties [23,28–30]. Several studies found that figs containing polyphenols and flavonoids played a significant role against diseases like diabetes, cancer, liver inflammation, skin disease, ulcers, and anemia [18,25,31]. Figs, as fresh fruit, are delicious, but fig-containing by-products are used due to the potential of food preservation, especially in cookie preparation and chocolate coating [32–34]. Moreover, figs containing anthocyanins played a significant role in preserving food colour and increasing the shelf-life of food [35].

However, fig fruit is a good source of bioactive compounds, but it has not yet been used in fish-based products. No studies have been performed to compare the effects of fig powder with BHT and AA on the quality of fresh fish balls. Our study aimed to investigate the quality of fish balls made from fig powder and compare their potential with BHT and AA by assessing the physicochemical test and their antioxidant potential.

## 2. Materials and methods

### 2.1 Preparation of fig powder

Fresh figs were procured from a nearby retail market. Following washing, the fig fruits were sliced into small fragments with a sharp kitchen knife. Subsequently, it was dried in an air oven dryer (Faithful, 101−3AB, Beijing, China) at 50°C for 48 hours and then minced to a finely ground powder using a household grinder. The particles were subjected to a 100-mesh sieve, wrapped in polythene bags, and maintained at 20°C until further utilization (Fig 1).

### 2.2 Preparation of fish ball

The experiment was not conducted on humans or animals. The pangasius catfish were purchased from a regional fish market in Jashore, Bangladesh. The fish went through deboning and was cut into small segments. Fish fat and lean fish meat were separated and ground twice individually with a meat grinder equipped with an eight mm-diameter plate with holes. Following grinding, the lean fish flesh and fish fat were carefully packed and stored at 4°C until subsequent

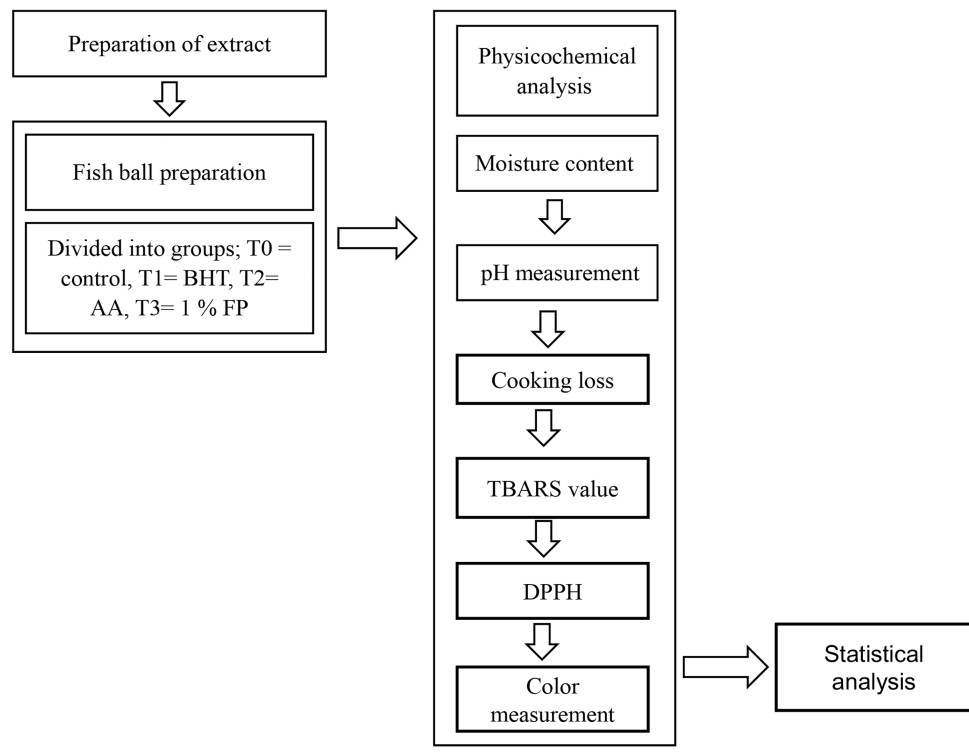

**Fig 1. Flowchart of the methodology.**

utilization. The lean fish flesh, fat, and all the unprocessed ingredients were thoroughly combined in the correct proportions as shown in Table 1. The fundamental components of the patties were as follows: 1.2% salt, 5.8% fat, 88% fish lean flesh, and 5% corn starch. Following this, three antioxidants fig powder, BHT, and AA were introduced one at a time to an equivalent portion of the meat mixture, and each addition was meticulously combined. Based on the subsequent formation, four types of fish balls were created: control, with 0.02% BHT (T1), 0.05% AA (T2), and 1% fig powder (T3). The fish ball had an approximate mass of 35 grams. For each sample, triplicate samples were produced. In total 48 fish balls were prepared. After the preparation, the fish balls were sealed in polythene containers and frozen at −20°C for 0, 1, and 2 months.

## 2.3 Physicochemical analysis

**2.3.1 Moisture content.** The moisture content of the fish ball sample was determined using Thiex's method (2009). The empty, clean Petri dishes were dried in a dry oven (Ecocell, MMM Group, Germany) at 105°C for 2 hours before being transferred to a desiccator to cool and then weighed (Acculab, TAWBA-5003, Germany) using 5g samples from each fish ball, handled with the utmost care. Afterwards, the weighted samples were placed onto the Petri dish and then in the oven dryer at 105°C for 4 hours. After drying, the petri dish was transferred to the desiccator to cool, and the samples were reweighed. The moisture content was calculated using the following formula [36].

$$\text{Moisture Content (\%)} = \frac{Weight\ of\ sample\ before\ drying - Weight\ of\ sample\ after\ drying}{Weight\ of\ sample\ before\ drying} \times 100$$

**2.3.2 PH measurement.** The pH was determined using a pH meter (Hanna, HI2211−2, Romania) coupled with an electrode. Using a Polytron homogenizer, 3g of prepared fish balls were homogenised for 30 seconds after being combined with 27 mL of distilled water. At 20 °C, two standard buffers with pH values of 4.0 and 7.0 were used to calibrate the pH meter. After obtaining three measurements for each sample, the average value of the measurements was computed [37].

**2.3.3 Cooking loss.** Previously published research articles suggested this study's method for measuring the cooking loss of all ready-to-cook fish balls. [38]. We calculated the weight difference between the fish balls before and after cooking. For this purpose, a water bath (BIOBASE, SY-2L4H, China) at 90°C for 30 minutes was used.

$$\text{Cooking loss (\%)} = (\frac{wt\ of\ fresh\ fish\ ball - wt.\ of\ cooked\ fish\ ball}{wt\ of\ fresh\ fish\ ball}) \times 100$$

**Table 1. Preparation of fish ball.**

| Ingredients (%) | Treatments | | | |
|---|---|---|---|---|
| | T0 | T1 | T2 | T3 |
| Fish Lean flesh | 88 | 88 | 88 | 88 |
| Fish Fat | 5.8 | 5.8 | 5.8 | 5.8 |
| Salt | 1.2 | 1.2 | 1.2 | 1.2 |
| Corn starch | 5 | 5 | 5 | 5 |
| Total | 100 | 100 | 100 | 100 |
| BHT | – | 0.02 | – | – |
| AA | – | – | 0.05 | – |
| Fig powder | – | – | – | 1 |

**2.3.4 TBARS measurement.** The modified method of TBARS was suggested by [39], to determine the degree of lipid oxidation in the ready-to-eat fish balls. The total amounts are indicated in milligrams of malondialdehyde (MDA) per kilogram of the sample. For this assay, 3 g of ground samples were subjected to high-speed homogenization with 27 mL of 3.86% perchloric acid via a digital homogenizer for 20 seconds. The mixture was then allowed to settle at a low temperature for one hour. Following a 10-minute centrifugation at 2000 rpm, the mixture was clarified with Whatman No. 1 filter paper to isolate the lipid component from the remaining macronutrients. 2 milliliters of filtered solution and 2 milliliters of 20-mM TBA solution were transferred to a test tube using a pipette. In another test tube, 2 milliliters of distilled water were added to 2 milliliters of 20-mM TBA solution to create a blank sample. After that, each solution was stored for 15 hours at room temperature with extreme care. In conclusion, the TBARS concentration was determined using a spectrophotometer (Dynamica HALO DB-20 Spectrophotometer, Australia), with an absorbance measurement obtained at 531 nm. For analysis, each sample's absorbance was measured in triplicate.

**2.3.5 DPPH measurement.** The radical scavenging activity of DPPH was determined using a modified version of the method suggested by [40] To determine the scavenging activity of different fish balls. After adding 2,850 µL of the methanol-based 24 mM DPPH solution to 150 µL of the sample, the mixture was left in a dark place for 24 hours. The absorbance measurement was conducted using a spectrophotometer (Dynamica HALO DB-20 Spectrophotometer, Australia) at a wavelength of 515 nm. DPPH radical scavenging activity was determined by calculating the percentage difference (%) in the sample absorbance.

$$\text{DPPH radical scavenging activity (\%)} = \frac{\text{Absorbance of control} - \text{Absorbance of sample}}{\text{Absorbance of control}} \times 100$$

**2.3.6 Colour measurement.** The assessment of colour in fresh fish balls was conducted by measuring the International Commission on Illumination lightness (L*), redness (a*), and yellowness (b*) using a Biobase BCM-110 colourimeter (Biobase, China) [41]. The instrument was calibrated using a standard white plate (Y = 82.2; x = 0.3151; y = 0.3263) to take a random triplicate measurement of each fish ball. The chroma value (C*) and hue angle (h°) were calculated using the internationally recognized equations. (1) and (2), respectively, are suggested by

$$\text{Chroma value} = \sqrt{a^{*2} + b^{*2}} \tag{1}$$

$$\text{Hue angle} = \tan^{-1}(b^*/a^*) \tag{2}$$

## 2.4 Statistical analysis

This study's analysis was conducted using IBM SPSS 26, and simple logistic regression graphs were visualized using GraphPad Prism software. Statistical analysis was conducted for each experiment-based (four treatments × three replications × three storage times) (S1 Table). Data collected from this study were presented as mean values of three replications with P-values (p < 0.05, significant) and standard error of the mean. The mean values of different variables were calculated using one-way ANOVA analysis. Storage periods at specific intervals and other treatments were used as variables. Post hoc analysis was also conducted to identify the significant relationships and differences among the samples.

## 3. Results and discussion

### 3.1 Effects of different types of antioxidants on the moisture percentage of fresh fish balls in frozen storage

The determination of moisture loss in different antioxidant-treated fish balls is presented in Table 2. Moisture content significantly (p < 0.05) decreased in all treated and non-treated samples. Control samples showed significantly (p < 0.05) the

**Table 2. Effects of different types of antioxidants on moisture percentage of fresh fish balls in frozen storage.**

| Months | T0 | T1 | T2 | T3 | P-value | SEM |
|---|---|---|---|---|---|---|
| 0 | 65.70[Ab] | 69.47[Aa] | 68.7[Aa] | 69.25[Aa] | 0.002 | 0.261 |
| 1 | 63.51[Bb] | 65.8[Bb] | 66.82[Bb] | 67.21[Bb] | 0.014 | 1.09 |
| 2 | 58.39[Cc] | 63.51[Cb] | 62.35[Cb] | 65.06[Ca] | 0.000 | 0.47 |
| P-value | 0.002 | 0.000 | 0.003 | 0.000 | | |
| SEM | 0.402 | 0.327 | 0.162 | 0.287 | | |

[a-d] Mean values in the same row with different letters demonstrated a significant difference ($p < 0.05$).

[A-C] Mean values in the same column with different letters demonstrated significant differences ($p < 0.05$).

T0: Control; T1: added 0.02% BHT; T2: added 0.05% Ascorbic acid; T3: added 1% Fig powder.

lowest moisture content, while FP retained significantly ($p < 0.05$) higher moisture levels. Initially, T1 samples had significantly ($p < 0.05$) the highest moisture content, which was significantly ($p < 0.05$) different from T0 samples. After 1 month of storage, no significant ($p > 0.05$) differences were observed among the samples. After 2 months, T3 and T0 samples exhibited significantly ($p < 0.05$) the highest and lowest moisture percentages, respectively. Both differed significantly ($p < 0.05$) from the other treated T1 and T2 samples.

Our result was followed by [14], demonstrating the gradual increase of moisture content in fish balls prepared with a higher percentage of moringa leaf powder. Similarly, other studies [42,43]. It was revealed that catfish cutlets and *Abalistes stellaris* fish balls prepared with moringa leaves powder had a higher moisture content than control samples, which is consistent with our present study. Therefore, fish balls prepared with fig powder had the best quality because they retained the moisture content compared to other samples.

### 3.2 Effects of different types of antioxidants on the pH of fresh fish balls in frozen storage

The pH value generally describes the acidity and alkalinity levels, which indicate the freshness of fish and meat-derived products [44]. The pH values of fresh fish balls treated with different antioxidant sources are presented along with control samples in Table 3. The pH values of all treated and non-treated samples increased significantly ($p < 0.05$) except in T3 samples. T1 samples had significantly ($p < 0.05$) the lowest pH values at the initial storage period; in contrast, T3 showed the highest ($p < 0.05$) values. After 1 month of frozen storage, it was found that T3 samples exhibited significantly ($p < 0.05$) the highest pH values, and T2 had the lowest values; a significant ($p < 0.05$) difference existed between T3 and other samples. At the end of the storage period, T0 samples had significantly the highest pH values, while T2 had the lowest. A non-significant ($p > 0.05$) difference between T2 and T3 samples and FP-treated samples also indicated a non-significant ($p > 0.05$) increase from months 1 to 2.

**Table 3. Effects of different types of antioxidants on the pH of fresh fish balls in frozen storage.**

| Month | T0 | T1 | T2 | T2 | P-value | SEM |
|---|---|---|---|---|---|---|
| 0 | 5.14[Cb] | 5.07[Cb] | 5.06[Cb] | 5.25[Ba] | 0.011 | 0.025 |
| 1 | 5.35[Bb] | 5.23[Bc] | 5.14[Bc] | 5.45[Aa] | 0.003 | 0.028 |
| 2 | 5.60[Aa] | 5.53[Aa] | 5.27[Ab] | 5.38[Ab] | 0.003 | 0.040 |
| P-value | 0.001 | 0.002 | 0.001 | 0.028 | | |
| SEM | 0.029 | 0.039 | 0.020 | 0.036 | | |

[a-d] Mean values in the same row with different letters represented a significant difference ($p < 0.05$).

[A-C] Mean values in the same column with different letters represent significant differences ($p < 0.05$).

T0: Control; T1: added 0.02% BHT; T2: added 0.05% Ascorbic acid; T3: added 1% Fig powder.

Some previous studies [18,45], demonstrated that mango peel powder and moringa extract reduced the pH value of fish products, which was consistent with our study. During the storage period, the spoilage bacteria could change the physicochemical attributes and increase the accumulation of ammonia and trimethylamine, increasing the pH value [46]. So, fig powder containing organic acid helps reduce the pH values of fish balls. Fig powder containing antioxidants and organic acid helps maintain its pH value for a longer time period in the frozen storage condition.

### 3.3 Effects of different types of antioxidants on the cooking loss of fresh fish balls in frozen storage

The cooking loss demonstrated moisture and fat loss after cooking the fish and meat products [47]. According to Table 4, non-significant difference was found between the control and treated samples. After the completion of frozen storage, T0 samples exhibited the highest cooking loss, and the difference was significant ($p < 0.05$). Our study found that cooking loss increased significantly in T0 samples ($p < 0.05$) with increasing time, whilst the T1, T2, and T3 samples increased non-significantly ($p > 0.05$), which indicated that BHT, AA, and FP were able to lower moisture and fat.

The results of this study were in agreement with previous studies conducted [19,48], in which they found that beef patties and chicken nuggets treated with clove extract and plum pulp powder showed a non-significant enhancement in cooking loss. Since cooking loss played a significant role in maintaining nutrition status, tenderness, and juiciness [49]. Fish balls treated with FP containing antioxidants were less affected during the storage period due to lower cooking loss.

### 3.4 Effects of different types of antioxidants on TBARS (mg MDA/kg of sample) values of fresh fish balls in frozen storage

TBARS value is a vital indicator of food quality. It is the principal indicator of lipid oxidation and rancidity in food products [50]. Changes in malondialdehyde concentration in fresh fish balls treated with different antioxidant sources are exhibited in Fig 2. At the beginning of the storage, T0 samples showed the significantly ($p < 0.05$) highest mg MDA/kg values, while T0 samples had the lowest significant ($p < 0.05$) values. The samples (control and treated) were significantly ($p < 0.05$) different from each other. After 1 month, T0 exhibited the significantly highest malondialdehyde (MDA) values; in contrast, T2 had the lowest level of MDA ($p < 0.05$). T0 samples were significantly difference from T1, T2, and T3, while T3 and T1 had nonsignificant difference ($p < 0.05$). At the end of the frozen storage, T2 had the lowest value ($p < 0.05$). At the same time, T1 and T3 had higher values than the T0 samples, and the difference was significant ($p < 0.05$). FP maintained significantly lower TBARS values, demonstrating its antioxidant potential by reducing oxidation.

A study reported that fish balls incorporated with thyme, basil, and rosemary exhibited the lowest TBARS values, which agreed with our present study [12]. Another study showed that fish fingers prepared from catfish using rosemary extracts positively reduced oxidation due to phenolic compounds, which was also consistent with our present study [51]. One more study identified that mangosteen peel-treated fish balls prepared from *Pangasius hypophthalmus fish* had comparatively

**Table 4. Effects of different types of antioxidants on the cooking loss of fresh fish balls in frozen storage.**

| Month | T0 (%) | T1 (%) | T2 (%) | T3 (%) | p-value | SEM |
|---|---|---|---|---|---|---|
| 0 | 13.65[Ca] | 13.45[Aa] | 13.54[Aa] | 13.49[Aa] | 0.047 | 0.0625 |
| 1 | 13.86[Ba] | 13.60[Aa] | 13.67[Aa] | 13.62[Aa] | 0.132 | 0.121 |
| 2 | 14.07[Aa] | 13.73[Ab] | 13.81[Ab] | 13.76[Ab] | 0.009 | 0.386 |
| p-value | 0.002 | 0.029 | 0.011 | 0.082 | | |
| SEM | 0.037 | 0.106 | 0.071 | 0.101 | | |

a-d Mean values in the same row with different letters represented a significant difference ($p < 0.05$).

A-C Mean values in the same column with different letters represent significant differences ($p < 0.05$).

T0: Control; T1: added 0.02% BHT; T2: added 0.05% Ascorbic acid; T3: added 1% Fig powder.

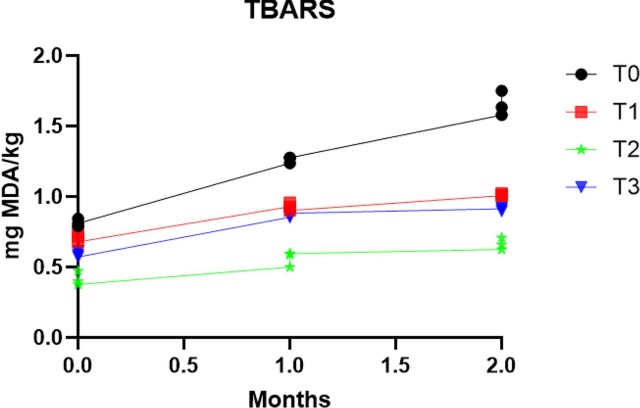

**Fig 2. TBARS (mg MDA/kg) values of different antioxidants added to fish balls in frozen condition.**

lower TBARS values than those treated with BHT [18]. Our study indicated that fig had the potential to enhance shelf life by reducing the oxidation of fish balls.

### 3.5 Effects of different types of antioxidants on the scavenging activity (%) of fresh fish balls in frozen storage

Food shelf-life and the antioxidant potential of natural preservatives are associated with DPPH values. DPPH is a measure that extensively demonstrates the antioxidant potential by assessing free radical scavenging activity [52]. Table 5 showed how the DPPH free radical scavenging activity changed over time in the various fish balls in frozen storage. Throughout the study, the antioxidant potentiality of T0 and T1 samples decreased significantly ($p<0.05$), while T2 and T3 samples did not change significantly after one month of storage. In the initial period, T2 exhibited the significantly highest antioxidant activity (52.1%), while the T0 sample exhibited the lowest ($p<0.05$). After 1 and 2 months of storage, T2 and T3 samples showed the highest free radical scavenging activity, while T0 showed the lowest free radical scavenging activity. The T2 and T3 samples significantly ($p<0.05$) differed from the T0 samples, and no significant difference was found among the T1, T2, and T3 samples. FP showed its activity as a potential source of antioxidants. Our study's free radical scavenging activity was consistent with [53–55] which found that black tea powder, rosemary, thyme, and mango peel powder had the highest scavenging activity. Another study demonstrated that Ficus carica had higher free radical scavenging activity than TBHQ when used to preserve the oxidative stability of canola oil [56]. Our assessment indicated that figs containing bioactive compounds reduce the free radical activity of fish balls, which was one possible reason for their higher antioxidant potential than others.

**Table 5. Effects of different types of antioxidants on the scavenging activity (%) of fresh fish balls in frozen storage.**

| Month | T0 (%) | T1 (%) | T2 (%) | T3 (%) | P-value | SEM |
|---|---|---|---|---|---|---|
| 0 | 41.37[Ac] | 52.12[Aa] | 51.90[Aa] | 51.07[Ab] | 0.000 | 0.417 |
| 1 | 35.40[Bb] | 50.03[Bb] | 50.166[Aa] | 49.94[Aa] | 0.093 | 0.521 |
| 2 | 27[Cb] | 45.30[Ca] | 44.94[Ba] | 47.07[Ba] | 0.000 | 0.941 |
| P-value | 0.000 | 0.000 | 0.000 | 0.000 | | |
| SEM | 0.795 | 0.296 | 0.95 | 0.382 | | |

[a-d] Mean values in the same row with different letters show a significant difference ($p<0.05$).

[A-C] Mean values in the same column with different letters exhibited significant differences ($p<0.05$).

T0: Control; T1: added 0.02% BHT; T2: added 0.05% Ascorbic acid; T3: added 1% Fig powder.

## 3.6 Effects of different types of antioxidants on the colour values of fresh fish balls in frozen storage

Consumer acceptability of food is associated with many factors; color is one of those parameters. The colour, including lightness, redness, and yellowness, values of fish and meat products significantly determine consumer acceptability [57]. Oxidation of lipids results in colour changes and undermines inherent pigments in meat [58]. Other possible factors responsible for colour changes are the incorporation of non-meat components, including fillers and meat extenders [50]. The fish ball's colour value, prepared with or without different food additives, is shown in Fig 3. Regarding lightness, T0 samples significantly showed the highest lightness values, while T3 indicated the lowest ($p<0.05$) throughout the frozen storage period. At the initial point, T0 was significantly ($p<0.05$) different from T2 and T3. After 1 month of storage, it was found that there was a significant ($p<0.05$) difference among the samples. After completing the frozen storage period, T0 showed the significantly ($p<0.05$) highest values, and it significantly ($p<0.05$) differed from synthetic and natural antioxidant-treated samples, whilst a non-significant ($p>0.05$) difference existed between T1 and T2 samples. A previous study showed that fish balls prepared with different concentrations of tea powder had lower lightness values than those of control samples [53], which was consistent with our study. Another study which also resulted in similar values to our study, demonstrated that fish products prepared with the addition of caffeic acid comparatively lower the lightness values than those of control and wheat dietary fiber-treated samples [59]. One more study conducted identified that rainbow trout croquettes prepared with the brewing dill extract had comparatively lower lightness values that was also in agreement with our present study [18]. Regarding redness (a*), values decreased in all samples significantly ($p<0.05$) with increasing time period. At month 0, T2 and T0 showed significantly ($p<0.05$) the highest and lowest a* values, respectively, and a significant ($p<0.05$) difference was found among the samples. After the completion of one month, T2 exhibited the highest (a*) values; it was significantly ($p<0.05$) different from T0, T1, and T3, whereas there was no significant ($p>0.05$) difference between T1 and T3. After the frozen storage period, T0 and T2 showed the significantly ($p<0.05$) lowest and highest redness values, respectively. T2 and T3 did not differ significantly ($p<0.05$). During this

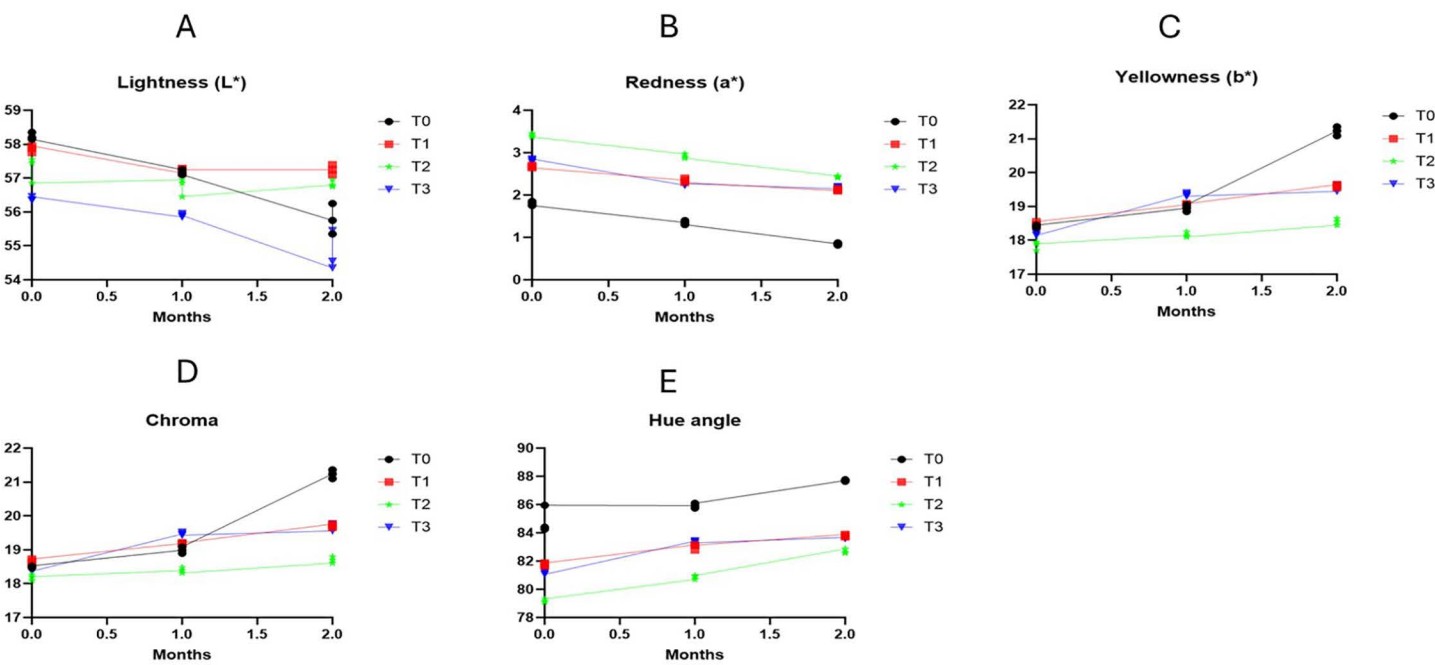

**Fig 3. Color values of fish balls under frozen conditions: (A) Lightness values; (B) Redness values; (C) Yellowness values; (D) Chroma values; (E) Hue angle.**

time, FP retained significantly higher redness values than control samples. Previous studies reported that meat and fish-based products prepared using fruit by-products retain higher redness values than control samples, consistent with our study [53,60,61]. A study was consistent with our study, which demonstrated rainbow trout croquettes prepared with the addition of brewing dill extract had the highest redness values, while the control sample had the lowest values. Yellowness (b*), chroma, and Hue values increased significantly over the period [18]. During the initial period and after one month, T1 samples had the highest b* values, which did not significantly differ from T0 samples but significantly differed from T2 and T3 ($p < 0.05$). After the storage period of 2 months, T0 samples had the highest b* values, while T2 samples had the lowest. T1 and T3 varied insignificantly, while T0 had a significant ($p < 0.05$) variance from other treated samples. The yellowness (b*) values obtained in this investigation were consistent with previous articles who reported that *Origanum vulgare* extract and clove extract-treated meat products had lower b* values than control samples [62,63]. Chroma and hue values were comparatively lower in all treated samples after the frozen storage period. At the beginning of the storage period, T3 samples' chroma value was lowest, and T1 had the highest values, but the difference was not significant. After a 2-month storage period, T2 had the significantly lowest chroma values, while T0 had the significantly highest values ($p < 0.05$). In terms of Hue angle (h°), control samples exhibited significantly the highest values, and T2 had the lowest values, respectively. In both Chroma and Hue angles, T3 samples had significantly ($p < 0.05$) different values from control samples, but there was no significant difference between T1 and T3 samples. A study found that beef patties treated with BHT, AA, and clove extract had significantly lower chroma values than those of control samples, and the hue values were lowest among the treated and control samples, which was consistent with our study [39]. Moreover, another study found that precooked beef meatballs prepared with the addition of nutmeg had the lowest chroma values [64]. The lower hue angle was lower than the control samples but higher than the BHT-treated samples. Our graph indicates that BHT, AA, and FP retained significantly lower hue values than the control samples. FP-treated samples retained colour values due to bioactive properties, which delayed the oxidation of fish and maintained color throughout the storage period.

## 4. Conclusion

The addition of BHT, AA, and fig powder significantly influenced the physicochemical properties of fish balls by maintaining moisture content, decreasing pH levels, and minimizing cooking loss after two months of frozen storage. After adding fig powder as antioxidants to the fish balls, the color of the fig-added samples was stabilized, especially when redness values were higher and yellowness values were lower than those of control samples. Adding 1% fig powder reduced the oxidation of fish balls by showing the highest scavenging activity, but the difference was not statistically significant with BHT and AA incorporated samples. The addition of antioxidants lowers the amount of malondialdehyde (mg MDA/kg) in all treated samples. Fig showed higher antioxidant potential than BHT by inhibiting lipid oxidation and reducing free radical activity. Therefore, it can be assumed that figs, as a potential source of antioxidants, could be used instead of synthetic antioxidants. In addition to our study, further research could be done on antimicrobial activity, protein oxidation, and heme iron tests.

## Supporting information

**S1 Table. Raw data of this research work.**
(XLSX)

## Acknowledgments

The authors show their gratitude to the Nutrition and Food Technology Department of Jashore University of Science and Technology for their assistance and support in the laboratory.

## Author contributions

**Conceptualization:** Md. Ashrafuzzaman Zahid.

**Data curation:** Md. Sakib Hasan.

**Formal analysis:** Md. Sakib Hasan, Md. Nowshad Mahmud Choyon, Md. Golam Rabby, Ananya Raiyan, Md. Mohaiminul Islam, Nishat Tabassum Prokrite, Md. Abid Hassan, Nishat Chadni Liza.

**Funding acquisition:** Md. Ashrafuzzaman Zahid.

**Supervision:** Md. Ashrafuzzaman Zahid.

**Writing – original draft:** Md. Sakib Hasan, Md. Golam Rabby, Suvasish Das Shuvo.

**Writing – review & editing:** Md. Numan Islam, Rashida Parvin, Md. Ashrafuzzaman Zahid.

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
