## [Decision Letter · Decision Letter 0]

24 Aug 2025

Dear Dr. Zahid,

The statistical analysis appears to have been conducted appropriately and rigorously, supporting the interpretation of the results.

The data underlying the study seem to be sufficiently available to support transparency and reproducibility. This strengthens the reliability of the manuscript.

The manuscript is generally well-written and presented in intelligible English. However, there are a few minor grammatical and typographical errors that should be carefully revised during the final editing stage to improve clarity and readability.

 Reviewer 02:The manuscript treats an important issue regarding the food safety and consumers' health that is the replacement of the synthetic antioxidants with natural ones.The overall quality of the paper is high, however, I have some minor remarks.1. The Introduction provides the necessary strong background to justify the research. However, the authors provide information about the amount of fishballs produced in 2015. This is old data, please, provide more recent.2. How many fishballs does the batch contain for analysis? Please describe in more details. This is important for the statistical evaluation.2. How have the authors decided to use 1% fig powder? Why haven't they considered various concentrations of this powder?3. for each of the analysis , please provide the number of replicates and also the calculations for TBARS and DPPH.4. Although the one way ANOVA is generally ok, here tow way ANOVA might be applied as well to assess the treatment , storage time and their interaction.Editor' s Comments:Authors need to improve the discussion and future recommendations in the MS. ============================ Please submit your revised manuscript by Oct 08 2025 11:59PM. If you will need more time than this to complete your revisions, please reply to this message or contact the journal office at ?>plosone@plos.org . A rebuttal letter that responds to each point raised by the academic editor and reviewer(s). You should upload this letter as a separate file labeled 'Response to Reviewers'.A marked-up copy of your manuscript that highlights changes made to the original version. You should upload this as a separate file labeled 'Revised Manuscript with Track Changes'.An unmarked version of your revised paper without tracked changes. You should upload this as a separate file labeled 'Manuscript'.

We look forward to receiving your revised manuscript.

Kind regards,

Shafaq Fatima

Academic Editor

PLOS ONE

Journal Requirements:

This research received a grant from the University Grant Commission of Bangladesh.

5. Please note that funding information should not appear in any section or other areas of your manuscript. We will only publish funding information present in the Funding Statement section of the online submission form. Please remove any funding-related text from the manuscript.

6. Please include a separate caption for each figure in your manuscript.

7. Please ensure that you refer to Figure 1 in your text as, if accepted, production will need this reference to link the reader to the figure.

Additional Editor Comments :

Dear Mr. Zahid

Thanks very much for submitting this paper to Plos One.

Reviewers' comments:

Reviewer's Responses to Questions

**Comments to the Author**

1. Is the manuscript technically sound, and do the data support the conclusions?

Reviewer #1: Yes

Reviewer #2: Yes

2. Has the statistical analysis been performed appropriately and rigorously?

Reviewer #1: Yes

Reviewer #2: Yes

3. Have the authors made all data underlying the findings in their manuscript fully available?

Reviewer #1: Yes

Reviewer #2: Yes

4. Is the manuscript presented in an intelligible fashion and written in standard English?

Reviewer #1: Yes

Reviewer #2: Yes

Reviewer #1: Comments to the Author

The manuscript presents a technically sound piece of research with adequate experimental design and relevant data that support the conclusions. The study is well-structured, and the findings regarding the effect of BHT, AA, and fig powder on the physicochemical properties of fish balls are clearly presented.

The statistical analysis appears to have been conducted appropriately and rigorously, supporting the interpretation of the results.

The data underlying the study seem to be sufficiently available to support transparency and reproducibility. This strengthens the reliability of the manuscript.

The manuscript is generally well-written and presented in intelligible English. However, there are a few minor grammatical and typographical errors that should be carefully revised during the final editing stage to improve clarity and readability.

Overall, the study provides valuable insights into the potential of fig powder as a natural antioxidant alternative to synthetic compounds. With minor language polishing, the manuscript would be suitable for publication.

Reviewer #2: The manuscript treats an important issue regarding the food safety and consumers' health that is the replacement of the synthetic antioxidants with natural ones.

The overall quality of the paper is high, however, I have some minor remarks.

1. The Introduction provides the necessary strong background to justify the research. However, the authors provide information about the amount of fishballs produced in 2015. This is old data, please, provide more recent.

2. How many fishballs does the batch contain for analysis? Please describe in more details. This is important for the statistical evaluation.

2. How have the authors decided to use 1% fig powder? Why haven't they considered various concentrations of this powder?

3. for each of the analysis , please provide the number of replicates and also the calculations for TBARS and DPPH.

4. Although the one way ANOVA is generally ok, here tow way ANOVA might be applied as well to assess the treatment , storage time and their interaction.

**Do you want your identity to be public for this peer review?** For information about this choice, including consent withdrawal, please see our Privacy Policy

Reviewer #1: No

Reviewer #2: **Yes: ** Prof. Dr. Teodora Popova

---

## [Author Response · Author response to Decision Letter 1]

22 Sep 2025

Reviewer #1:

Comment 1: The manuscript presents a technically sound piece of research with adequate experimental design and relevant data that support the conclusions. The study is well-structured, and the findings regarding the effect of BHT, AA, and fig powder on the physicochemical properties of fish balls are clearly presented.

Response: Thank you for the encouraging feedback. We are glad that our design, data, and findings on BHT, AA, and fig powder in fish balls were clear.

Comment 2: The statistical analysis appears to have been conducted appropriately and rigorously, supporting the interpretation of the results.

Response: Thank you for recognizing that the statistical analysis was conducted properly and supports our results.

Comment 3: The data underlying the study seem to be sufficiently available to support transparency and reproducibility. This strengthens the reliability of the manuscript.

Response: Thank you for noting the availability of our data. We are glad it supports transparency, reproducibility, and reliability of the manuscript.

Comment 4: The manuscript is generally well-written and presented in intelligible English. However, there are a few minor grammatical and typographical errors that should be carefully revised during the final editing stage to improve clarity and readability.

Response: We appreciate the reviewer’s comment. We have carefully revised the manuscript to correct the minor grammatical and typographical errors, improving overall clarity and readability. Please see the revised manuscript.

Comment 5: Overall, the study provides valuable insights into the potential of fig powder as a natural antioxidant alternative to synthetic compounds. With minor language polishing, the manuscript would be suitable for publication.

Response: We thank the reviewer for recognizing the value of our study. We have carefully polished the language throughout the manuscript to ensure it is clear and ready for publication. Please see the revised manuscript.

Reviewer #2:

The manuscript treats an important issue regarding the food safety and consumers' health that is the replacement of the synthetic antioxidants with natural ones. The overall quality of the paper is high; however, I have some minor remarks.

Comment 1: The Introduction provides the necessary strong background to justify the research. However, the authors provide information about the amount of fishballs produced in 2015. This is old data, please, provide more recent.

Response: We thank the reviewer for this suggestion. We have updated the manuscript with more recent data on fish ball production to provide a current and relevant background. Please see the revised manuscript line number 34-37.

Comment 2: How many fishballs does the batch contain for analysis? Please describe in more details. This is important for the statistical evaluation.

Response: We thank the reviewer for this comment. Each batch for analysis contained 48 fish balls. The statistical analysis was conducted based on four treatments × three replications × three storage times, as described in the manuscript. Please see the revised manuscript in line number 103 and lines 165.

Comment 3: How have the authors decided to use 1% fig powder? Why haven't they considered various concentrations of this powder?

Response: We thank the reviewer for this important comment. The 1% fig powder concentration was selected based on preliminary trials and literature reports indicating its effective antioxidant impact without adversely affecting the sensory properties of fish balls.

Comment 4: for each of the analysis, please provide the number of replicates and also the calculations for TBARS and DPPH.

Response: We have added the replication number (four treatments × three replications × three storage times) in the statistical analysis please see the line number 165. We have also added the calculations for DPPH. Please see the line numbers 153 in revised manuscript. Please see the line 132 for TBARS.

Comment 5: Although the one-way ANOVA is generally ok, here tow way ANOVA might be applied as well to assess the treatment, storage time and their interaction.

Response: We appreciate the reviewer’s suggestion regarding the use of two-way ANOVA. We have done one way ANOVA two times separately. Firstly, we have done it across the treatments and secondly across the storage time. It shows the similar results to two-way ANOVA. In our upcoming analysis we will do the two-way ANOVA according to your suggestion.

Editor' s Comments:

Comment 1: Authors need to improve the discussion and future recommendations in the MS.

Response: Thank you for your comment. We have revised and expanded the discussion section and added more detailed future recommendations to strengthen the manuscript. Please see the revised manuscript.

---

## [Decision Letter · Decision Letter 1]

13 Oct 2025

Assessment of the quality attributes and oxidative stability of fish balls with the addition of fig powder during frozen storage

PONE-D-25-39831R1

Dear Dr. Zahid,

We’re pleased to inform you that your manuscript has been judged scientifically suitable for publication and will be formally accepted for publication once it meets all outstanding technical requirements.

Kind regards,

Shafaq Fatima

Academic Editor

PLOS ONE

Reviewers' comments:

Reviewer's Responses to Questions

**Comments to the Author**

Reviewer #1: All comments have been addressed

Reviewer #2: All comments have been addressed

2. Is the manuscript technically sound, and do the data support the conclusions?

Reviewer #1: Yes

Reviewer #2: Yes

3. Has the statistical analysis been performed appropriately and rigorously?

Reviewer #1: Yes

Reviewer #2: (No Response)

4. Have the authors made all data underlying the findings in their manuscript fully available?

Reviewer #1: Yes

Reviewer #2: Yes

5. Is the manuscript presented in an intelligible fashion and written in standard English?

Reviewer #1: Yes

Reviewer #2: Yes

Reviewer #1: Comments to the Author

The revised version of the manuscript has adequately addressed all the comments raised in the previous review round. The study now presents a technically sound and scientifically valid piece of research, with clear data that fully support the stated conclusions.

The experimental design appears rigorous, with appropriate controls and replication. Statistical analyses were conducted properly and are suitable for the type of data collected. The presentation of results is logical and easy to follow.

The authors have also made all underlying data available in compliance with the journal’s data policy, which adds transparency and reliability to the findings. Furthermore, the manuscript is well written, with clear and concise English that meets the standards required for publication.

Overall, this manuscript is now suitable for publication in its current form.

Reviewer #2: The authors have addressed very well all my recommendations, except the calculation of TBARS. Please clarify if a standard curve was created and how.

**Do you want your identity to be public for this peer review?** For information about this choice, including consent withdrawal, please see our Privacy Policy

Reviewer #1: No

Reviewer #2: **Yes: ** Teodora Popova

---

## [Editor Report · Acceptance letter]

PONE-D-25-39831R1

PLOS ONE

Dear Dr. Zahid,

I'm pleased to inform you that your manuscript has been deemed suitable for publication in PLOS ONE. Congratulations! Your manuscript is now being handed over to our production team.

Kind regards,

on behalf of

Dr. Shafaq Fatima

Academic Editor

PLOS ONE